# Characterization of *Delonix regia* Flowers’ Pigment and Polysaccharides: Evaluating Their Antibacterial, Anticancer, and Antioxidant Activities and Their Application as a Natural Colorant and Sweetener in Beverages

**DOI:** 10.3390/molecules28073243

**Published:** 2023-04-05

**Authors:** Doaa Ebada, Hefnawy T. Hefnawy, Ayman Gomaa, Amira M. Alghamdi, Asmaa Ali Alharbi, Mohammed S. Almuhayawi, Mohanned Talal Alharbi, Ahmed Awad, Soad K. Al Jaouni, Samy Selim, Gehad S. Eldeeb, Mohammad Namir

**Affiliations:** 1Biochemistry Department, Faculty of Agriculture, Zagazig University, Zagazig 44511, Egypt; hefnawytaha@gmail.com (H.T.H.); aymangoma@gmail.com (A.G.);; 2Department of Biochemistry, Faculty of Science, King Abdulaziz University, Jeddah 21589, Saudi Arabia; amira.alghmadi78@gmail.com (A.M.A.); aanalharbi@kau.edu.sa (A.A.A.); 3Department of Clinical Microbiology and Immunology, Faculty of Medicine, King Abdulaziz University, Jeddah 21589, Saudi Arabia; 4Department of Medical Microbiology and Parasitology, Faculty of Medicine, University of Jeddah, Jeddah 21589, Saudi Arabia; 5Department of Hematology/Oncology, Yousef Abdulatif Jameel Scientific Chair of Prophetic Medicine Application, Faculty of Medicine, King Abdulaziz University, Jeddah 21589, Saudi Arabia; 6Department of Clinical Laboratory Sciences, College of Applied Medical Sciences, Jouf University, Sakaka 72388, Saudi Arabia; 7Food Technology Department, Faculty of Agriculture, Suez Canal University, Ismailia 41511, Egypt; 8Department of Food Science, Faculty of Agriculture, Zagazig University, Zagazig 44511, Egypt

**Keywords:** *Delonix regia*, antioxidant, antimicrobial, resistance, bacterial infection, virulent, anthocyanins, polysaccharides, beverages, color, taste

## Abstract

In the present study, an attempt was made to investigate the in vitro antioxidant, anticancer, and antibacterial activities of *Delonix regia*, then in vivo evaluate its safety as a natural colorant and sweetener in beverages compared to synthetic colorant and sweetener in rats, then serve the beverages for sensory evaluation. *Delonix regia* flowers had high protein, polysaccharide, Ca, Na, Mg, K, and Fe contents. The *Delonix regia* pigment extract (DRPE) polysaccharides were separated and purified by gel permeation chromatography on Sephacryl S-200, characterized by rich polysaccharides (13.6 g/L). The HPLC sugar profile detected the monosaccharides in the extracted polysaccharides, composed of mannose, galactose, glucose, arabinose, and gluconic acid, and the structure of saccharides was confirmed by FTIR, which showed three active groups: carbonyl, hydrocarbon, and hydroxyl. On the other hand, the red pigment constituents of DRPE were detected by HPLC; the main compounds were delphinidin and cyanidin at 15 µg/mL. The DRPE contained a considerable amount (26.33 mg/g) of anthocyanins, phenolic compounds (64.7 mg/g), and flavonoids (10.30 mg/g), thus influencing the antioxidant activity of the DRPE, which scavenged 92% of DPPH free radicals. Additionally, it inhibited the population of pathogenic bacteria, including *Staphylococcus aureus*, *Listeria monocyogenes*, *Salmonella typhimurum*, and *Pseudomonas aeruginosa,* in the range of 30–90 μg/mL, in addition to inhibiting 85% of pancreatic cancer cell lines. On the in vivo level, the rats that were delivered a diet containing DRPE showed regular liver markers (AST, ALP, and ALT); kidney markers (urea and creatinine); high TP, TA, and GSH; and low MDA, while rats treated with synthetic dye and aspartame showed higher liver and kidney markers; lowered TP, TA, and GSH; and high MDA. After proving the safety of DRPE, it can be safely added to strawberry beverages. Significant sensorial traits, enhanced red color, and taste characterize the strawberry beverages supplemented with DRPE. The lightness and redness of strawberries were enhanced, and the color change ΔE values in DRPE-supplemented beverages ranged from 1.1 to 1.35 compared to 1.69 in controls, indicating the preservative role of DRPE on color. So, including DRPE in food formulation as a natural colorant and sweetener is recommended for preserving health and the environment.

## 1. Introduction

Most of the world’s population uses medicinal herbs as their sole form of medicine. Herbal medicines are already the foundation of therapeutic usage in developing nations, although there has been a recent growth in their use worldwide [1]. In recent decades, the therapeutic use of plants has gained greater significance because the extracts of medicinal plants include minerals, primary metabolites, and a wide variety of secondary metabolites with antioxidant properties [2].

Free radicals have been implicated in the etiology of several illnesses, including atherosclerosis, ischemic heart disease, cancer, Alzheimer’s and Parkinson’s diseases, liver and kidney disorders, and aging. Inflicting oxidative damage on lipids, proteins, and nucleic acids, cytotoxic active ROS can severely disturb normal metabolism [3,4].

Natural antioxidants indigenous to plant foods can scavenge RONS; research shows that they may significantly reduce the beginning of human oxidative illness [5].

Phenolic chemicals contribute significantly to the color and flavor of fruits and vegetables [6], in addition to participating in growth and reproduction activities and defending against diseases and predators. Hence, a plant with high antioxidants exhibits substantial resistance to ROS-induced oxidative damage [7].

The flowering plant species *Delonix regia* (Bojer ex Hook.) belongs to the Fabaceae family. It is known as Krisnachura in Bangladesh, the flame tree, because of its stunning red flowers. The tree is native to Madagascar and is commonly planted as a decorative tree in many tropical nations. The flower of this tree can treat many diseases, such as rheumatoid arthritis, constipation, pneumonia, inflammation, diabetes, and malaria. On the other hand, the leaf extracts are utilized as antibacterial, antioxidant, antihyperglycemic, anti-inflammatory, cytotoxic, and hepatoprotective agents. It was discovered that stem bark preparations contain antioxidants and have anticancer, hepatoprotective, cytotoxic, gastroprotective, and antimicrobial properties, as reported by Rahman et al. [8].

The flowers of *Delonix regia* are rich in polysaccharides and pigments. Polysaccharides are a significant bioactive material with a variety of physiological activities, including immunity, cell growth regulation, and senescence. They have a variety of medical applications, health goods, materials, and functional meals [9]. As a green bio-medical product, it possesses promising commercial potential. Plant polysaccharides have exhibited potential antioxidant activities, and we have reviewed some previously. These properties depend on the structure, such as the composition, linkage pattern, and sequence of the monosaccharide, the molecular conformation, and the stereo configuration [10]. Additionally, polysaccharides have health and industrial benefits; they are called prebiotics, which improve host health by stimulating the growth of gut microbiota; additionally, they are added to functional foods and studied in pharmaceutical and industrial research fields [11].

On the other hand, the flowers of *D. regia* are rich in carotenoids, particularly β-carotene, followed by significant levels of lycopene isomers and several epoxy carotenoids. Such advantageous substances enable the use of flowers as pro-vitamin A, a natural source of color, and an ingredient in cosmetics, feed, and meals. Furthermore, anthocyanins are widely dispersed in nature and display beautiful colors and many health-promoting benefits; hence, they are frequently utilized in the food, pharmaceutical, and cosmetic sectors [12]. The *D. regia* with red and white flowers indicates the presence of anthocyanins, which are generated from phenol-propaniods such as peonidin-3-O-glucoside, petunidin-3-O-acetyl glucoside, and cyanidin-3-O rutinoside. Anthocyanins are widely dispersed in nature and display beautiful colors and many health-promoting benefits; hence, they are frequently utilized in food, pharmaceuticals, and cosmetics. As red wine shows, a rich source of anthocyanins, dietary polyphenols, and anthocyanins have positive and constructive effects on the human body, such as reducing the risk of cardiovascular, liver, and kidney illnesses [13]. A high concentration of antioxidants increases the utility of all-natural ingredients in various goods. *D. regia* has become a significant source of antioxidants that inhibit the development of ROS in the body [14].

The long-term use of synthetic antioxidant agents has been reported to cause carcinogenesis and liver injury [15]. Hence, scientists are now searching for natural antioxidants in plant sources for food or medicinal materials as an alternative to synthetic antioxidants due to their safety as reported in many studies. Rahman et al. [16] found that flower extracts from *Delonix regia* were rich in phytochemicals and potential bioactivities, which indicates the possibility of these flowers being used as a source of phytochemicals as well as a safe and effective natural medicine. Furthermore, El-Sayed et al. [17] suggested using the ethanolic extract of the flowers of *D. regia* as a chemopreventive agent against the two main causes of liver damage: liver toxicity by chlorinated agents and liver cancer. Additionally, the alcoholic extract of the *Delonix regia* Linn flowers was evaluated for antiarthritic activity in adult female Wistar rats, where it safely reduced paw edema volume [18]. Moreover, Morsi et al. [19] found that the carotenoid-rich extract of DR flowers can be used as a bio-preservative in sunflower oil for stability against oxidation.

No previous studies have used *D. regia* flowers as additives in food formulation as colorants and sweeteners, so in this study, we separated and purified the *D. regia* pigments extract, characterized the constituents of polysaccharides and pigments by HPLC, FTIR, and NMR, then evaluated the activity of DRPE as an antioxidant, anticancer, and antibacterial; the safety of DRPE as a food additive was estimated on rats; and then tested the sensorial traits of the DRPE-supplemented beverage.

## 2. Results and Discussion

### 2.1. Proximate Composition of Delonix regia

The analyses of *Delonix regia* content are recorded in Table 1. The results showed that ash, crude protein, crude fat, crude fiber, carbohydrate, and moisture were valued at 2.05, 10.01, 13.68, 14.04, 51.22, and 10.14%, respectively. The results showed that the plant contained a lot of protein, fat, fiber, and carbohydrates. These results align with those obtained by Oyedeji et al. [20] on *Delonix regia* seeds, who found that the % proximate analysis was moisture (10.1), crude fiber (14.6), ash (1.03), crude fat (17.16), crude protein (8.7), and carbohydrate (48). The mineral contents determined by atomic absorption in the *Delonix regia* sample were higher in manganese, magnesium, zinc, cobalt, potassium, phosphorus, and calcium than those reported by Oyedeji et al. [20], with mean values less than 13.81, 2220, 671.9, 0.012, 16,358, 2.087, and 4617 ppm, respectively (Table 1).

### 2.2. Phenolic Compounds

#### 2.2.1. Total Phenols and Flavonoid Contents in DRPE

In Figure 1A,B, the phenolic compounds and flavonoids increased in a DRPE concentration-dependent manner, where their content in DRPE (800 µg/mL) was estimated at 64.7 and 10.3 mg/g, respectively, which increased to 110 and 36 mg/g at a concentration of 1800 µg/mL. Polyphenol molecules are an essential antioxidant component contributing to numerous plants’ therapeutic benefits [21]. Chhabra and Gupta [22] found that the phenolic compound content in DR flowers was about 101 mg/g; however, Bohrga et al. [23] found that flower extract contains a 30 mg/g gallic acid equivalent. The quantity of polyphenols in raw plant material is a consequence of several factors involving genotype, environmental circumstances, altitude, light, temperature, and the amount of nutritional material in the soil [24].

#### 2.2.2. Anthocyanin Content

Anthocyanins are widely dispersed in nature and display beautiful colors and many health-promoting benefits; hence, they are frequently utilized in food, pharmaceuticals, and cosmetics [12]. Figure 2 shows that DR flowers are rich in anthocyanin content, which increases with the concentration of DRPE. The DRPE contains 51.2 µg/g Cyn 3-O-glu at a 1800 µg/mL concentration. Chhabra and Gupta [22] found that TAs in 1 g of DR flowers were 5000 µg/L. Additionally, Hosny et al. [25] found that the anthocyanin content was 22 µg/mL. Detecting monomeric anthocyanin by the pH differential method is a well-recognized, quick technique for determining the anthocyanin concentration in a plant extract or juice. The approach assumes that monomeric anthocyanins have low or no absorbance at pH 4.5; however, polymeric anthocyanins have absorbance at this pH. Due to a shift in these molecules’ reversible or irreversible equilibrium structure, pH has the greatest impact on the color of anthocyanins in the solution.

To elucidate the anthocyanin profile in DRPE, the HPLC profile in Figure 3 shows that delphinidin and cyanidin were the main anthocyanin pigment contents in DRPE, with about 15 µg/mL. Vegias et al. [26] stated that the petals of *D. regia* are abundant in a unique mix of physiologically active carotenoids and anthocyanins, especially cyanidin-3-O-rutinoside (8 g/100 g DW), which holds tremendous potential for culinary and medicinal uses.

### 2.3. Isolation, Characterization, and Identification of Polysaccharide Contents in DRPE

#### 2.3.1. Isolation

After 5 days of extraction, the polysaccharide content peaked at 13.6 g/L of crude polysaccharide. The primary active polysaccharide fraction was purified using Sephacryl S-200 gel-filtration columns. A significant fraction was recovered and lyophilized for further structural characterization and different activities. The average molecular weight of polysaccharide was 2.7 × 10^5^ kDa, determined by the GPC technique, which correlated with Rodriguez-Canto et al. [27], who found galactomannans enzymatically isolated from DR have a molecular weight of 4.86 × 10^5^ to 1.95 × 10^5^ Da. Figure 4 shows the GPC profile of isolated polysaccharides as having a single, symmetrically sharp peak, indicating that they were homogenous. These levels were increased but were comparable to other galactomannans described in the previous studies. Bunge seed galactomannan has an Mw of 1.1 × 10^6^ and an Mn of 0.82 × 10^5^ g/mol, as determined by [28]. In addition, the Mn of river tamarind galactomannans was 7.23 × 10^5^ g/mol, according to [29]. It is widely known that the molecular weight of galactomannans can vary significantly based on geographical and botanical origins, extracting conditions (e.g., temperature, pH, and organic solvents), and degree of galactosyl replacement or depolymerization [30].

#### 2.3.2. Characterization

##### FTIR Spectra

Figure 5 shows the FTIR spectra of isolated polysaccharides. The characteristic polysaccharide spectral bands were between 527 and 3351 cm^−1^. The strong peak at 3351.0 cm^−1^ matched the hydroxyl group originating from polysaccharides and water; the bands at 2929.2 cm^−1^ were attributable to the hydrocarbon group. At 1636.3 cm^−1^, corresponding to C=O. According to Fernando et al. [31], the signals between 1222.1 and 1547.7 cm^−1^ correspond to C-OH rotating vibrations. FTIR bands between 871.1 and 811.5 cm^−1^ were attributable to α-glycoside linkages and α-D-galactopyranose residue, respectively. The FTIR bands at 1146.0 cm^−1^ were assigned to the carbonyl group of carbohydrates, but 1023.6 and 1068.3 cm^−1^ indicated a pyranose form for polysaccharide-forming carbohydrates. Jiang et al. [32].

##### ^1^ H-NMR Analysis of *Delonix regia* Polysaccharide

The polysaccharide obtained from *Delonix regia* polysaccharide was analyzed by ^1^ H NMR spectroscopy. Analysis of the polysaccharide revealed specific carbohydrate signals of galactomannans, confirming the monosaccharide’s composition (Figure 6). The 2 separate signals at 5.51 and 5.23 ppm correspond to the anomeric protons H-1 of α-D-galactopyranose (G1) and H-1 of β-D-mannopyranose (M1/M1’) in the 1 H NMR spectrum. Based on the integrations of the respective peak areas of these 2 signals, the molar ratio of M/G was estimated to be 1.96. The polysaccharide extracted from Delonix regia seeds in Nigeria consisted of mannose and galactose with a ratio of 5:1 and an average molar mass of 7.23 × 10⁵ g/mol [33]. This value was coherent with that obtained after HPLC analysis (M/G = 2.22). The structure of enzymatically hydrolyzed DR galactomannan, as detected by FT-IR and DRX, retains an amorphous nature with just a minor increase in the galactose:mannose ratio as reported by ^1^ H NMR [27]. On the other hand, Da Silva et al. [34] found that 1D and 2D NMR spectroscopy indicated that the *Cassia fistula* polysaccharides consist of 4-linked β-D-mannose units and galactose units as anticipating groups. These galactose units are in the center of the core through a 1–6 linkage, and the galactomannan displayed a man/gal ratio of 3.1/1.

Therefore, the FTIR and NMR spectra indicated that polysaccharides in *D. regia* are galactomannans.

#### 2.3.3. Identification of Sugar Profile of Isolated Galactomannans by HPLC

The purified polysaccharide, a white powder, was used for subsequent analysis. It had a negative response to the Bradford test and no absorption at 280 or 260 nm in the UV spectrum, indicating the absence of protein and nucleic acid. The monosaccharide profile of HPLC shown in Figure 7 and Table 2 indicated that the polysaccharide was composed of mannose, glucose, galactose, xylose, and arabinose with a molar concentration of 3.60, 1.00, 2.70, 0.78, and 1.10 µM, respectively. A HPLC profile found the following monosaccharides in tea: mannose, ribose, rhamnose, glucuronic and galacturonic acids, glucose, xylose, galactose, and arabinose, with corresponding molar concentrations of 0.72, 0.78, 0.89, 0.13, 0.15, 0.36, 0.39, 0.36, 0.36, and 0.38 µM [35].

### 2.4. Biological Activities of DRPE

#### 2.4.1. Antioxidant

##### DPPH Assay

In our study, DRPE has high levels of phenolic, flavonoid, and volatile compounds that are highly related to high levels of radical scavenging [36]. In Figure 8A, DRPE (800 µg/mL) significantly scavenged 70% of DPPH free radicals, which increased to 93.55% when the DRPE concentration was 1800 µg/mL, comparable with ascorbic acid. The SC_50_ of DRPE was 200 µg/mL. Our results are correlated with those of Chhabra and Gupta [22], who found that water extracts of DR flowers scavenged 93.74% of DPPH free radicals.

##### Ferric-Reducing Antioxidant Power (FRAP)

As shown in Figure 8B, DRPE showed higher ferric-reducing power when compared with ascorbic acid. It could be concluded that TPC and ferric-reducing power are related. Fe^+3^ reductions are usually used to indicate electron-donating potential, an essential mechanism of antioxidant activity. DRPE (1800 µg/mL) can reduce ferric ions into ferrous ions with an absorbance of 5 at 700 nm, compared to ascorbic acid with a relative increase of 20%. Sharma et al. [37] found that the reducing power capacity of an ethanolic extract of the green hull of *J. regia* (500 μg/mL) exhibited an absorbance of 5.2. At a concentration of 400 µg/mL, the absorbance of 700 nm was measured for ferric reduction of the ethanolic extract of *J. regia*, which is close to the standard BHT.

#### 2.4.2. Antibacterial

Traditional antimicrobial treatment using antibiotics has the primary disadvantage of fast resistance development to current antimicrobial drugs. Therefore, it is vital to create novel antimicrobial medicines to combat drug-resistant bacteria that are undergoing continual genetic mutation [38]. The WHO has revealed that >75% of the world’s population relies mainly on plant-based traditional remedies, particularly in emerging and underdeveloped nations. Based on the screening of plant extracts for antimicrobial activity, it has been determined that higher plants are a prospective source of novel antimicrobial agents [38]. DRPE has broad-spectrum antibacterial activity against tested pathogenic bacteria (Figure 9); the inhibition zone diameters (IZDs) indicate that. The IZDs increased with concentration, ranging from 10 to 41 mm, against tested bacteria, which excelled in antibiotic zones. SA was the most vulnerable bacteria to DRBE 1800 µg/mL (41 mm), while ST was more resistant (27 mm). Chhabra and Gupta [22] found that DR flower extract has antibacterial activity against *Shigella flexneri, S. epidermidis,* and *E. coli* in the IZDs range of 9–22 mm. Additionally, our results are similar to those of Aylate et al. (2017). who stated that *Codiaeum variegatum* extracts from the croton plant showed high antibacterial activity also stated that the extract showed the lowest inhibition zone (9.25 mm) against *Listeria monocytogenes* and the highest inhibition zone (21 mm) was seen against *Salmonella typhimurium* (15 mm) and *Pseudomonas aeruginosa* (20 mm). In addition, the extract was significantly better than the effect of tetracycline against *Staphylococcus aureus*.

On the other hand, Figure 10 shows the lowest DRPE concentration inhabiting the tested bacteria and fungi populations, ranging between 20 and 40 µg/mL. The MIC against SA was the lowest (20 µg/mL), and the highest against ST was 40 µg/mL.

#### 2.4.3. Anticancer

Figure 11 shows that LPE has significant anticancer activity against the pancreatic cancer population compared to doxorubicin. The viability of cancer cells (*p* < 0.05) increased in concentration dependence. DRPE (1600 µg/mL) inhibited the viability of MCF-7 cell lines by 82% compared to DOX with 79%, and that correlated with the microscopic image that shows the inhibitory effect is higher in DRPE than DOX at a concentration of 1600 µg/mL, which is a clear indication for mitigating the oxidative stress on human cells. Comparing the vitality of pancreatic cancer cells treated with *D. regia* extract containing AgNPs to that of control cells, the results demonstrated a significant decrease. MTT analysis revealed that the IC_50_ of *D. regia* extract on pancreatic cell lines was 0.5 mg/mL, while the maximum inhibition of cell lines was observed at 1.5 mg/L concentration [39].

### 2.5. Safety Experiment

Table 3 shows the serum parameters of rats that received some additives to their diet, synthetic (dye and aspartame) and natural (DRPE). Adding synthetic dye at 1.6 mg/g to rats’ diets significantly increased ALT, AST, and MDA by 97, 153, and 54%, respectively, compared to the control, while decreasing GSH by 34% compared to the control, indicating liver damage caused by the dye. On the other hand, urea and creatinine increased by 117 and 65%, respectively, indicating kidney damage caused by the dye. Similarly, the lipid profile (LDL, TG, and TC) increased by 443, 94, and 91%, while HDL decreased. The effect of aspartame’s (1.6 mg/g) addition to the diet significantly causes oxidative damage to the liver and kidney at the end of the experiment.

On the contrary, the addition of DRPE to rats’ diets significantly enhances the levels of GSH, HDL, urea, and creatinine to levels comparable to control while lowering the levels of LDL, TG, TC, ALT, and AST, maintaining the liver and kidney functions, so adding DRPE as a colorant and sweetener in food formulation is very safe. Lotfy et al. [40] All rats fed 5% *Delonix regia* pods, leaves, or a mixture exhibited a significant decrease in LDL, TG, LDL, kidney functions, VLDL, and liver enzymes but a significant increase in total protein and high-density lipoprotein. Additionally, histopathological examinations of the kidneys revealed an improvement. This study indicated that *Delonix regia* benefits the kidney health of rats with renal failure. In addition, it improved liver function, the lipid profile, and histological examination in rats with renal failure.

### 2.6. The Experiment of Strawberry Beverage Supplemented with DRPE at Different Concentrations (200, 800, and 1600 µg/g) as a Natural Colorant and Sweetener

#### Sensory Properties and Color Changes

Table 4 displays the sensory assessment of strawberry beverages enriched with three concentrations of DRP (200, 800, and 1600 µg/g) during cold storage at 4 °C for 30 days. The strawberry beverages were treated with varied amounts of DRP for four weeks. DRPE significantly preserved and enriched the red color of strawberry beverages compared to the control because of the high anthocyanin content in DR. The color parameters whiteness (*L*) and redness (*b*) were enhanced by DR addition, while the color change in DR strawberry ranged from 1.35–1.1 compared to control 1.7. The results revealed that the highest taste was obtained by adding 200 and 800 µg/g DR powder to a strawberry formulation, with no sense between the concentrations because of the high polysaccharide content in DR. Other sensory qualities differed considerably from the control. At a concentration of 1600 µg/g, all sensorial traits, such as color, shape, flavor, texture, and overall acceptability, partially deteriorated. The overall acceptability of DR-strawberry (200 and 800 µg/g) was 8.1 and 8.3, increasing with gradient decrements and storage length, reaching 8.0 after four weeks of storage for DRSB3 (1600 µg/g). When increasing DRP concentration, there are no significant differences.

All strawberry samples achieved the highest sensory scores at the beginning of preservation due to their enhanced color, taste intensity, and consistency. However, after 30 days, the acidity of SB increased, and their sensory scores gradually decreased. The overall quality of SB improved with storage for up to 25 days before deteriorating. It can thus be attributed to the production of acidity. Generally, SB with variable concentrations and levels of DR remained stable after storage. These findings can be applied to developing the color and taste of functional beverages with superior antimicrobial, antioxidant, and anticancer properties without compromising the sensory quality of strawberry beverages, thereby enhancing the color and taste quality of the final product and masking defects through the natural production of aroma, color, and taste.

## 3. Materials and Methods

### 3.1. Materials

The flowers of *Delonix regia* were collected at the end of June 2020 (the end of the flowering season) from the Faculty of Agriculture, Zagaizg University, Egypt. All chemicals and solvents in this study were of analytical grade. The bacterial strains *Staphylococcus aureus*, *Listeria monocyogenes*, *Salmonella typhimurum*, and *Pseudomonas aeruginosa* were used in this study to determine the antimicrobial activity of DRPE.

### 3.2. Proximate Analysis of Delonix regia Flowers

Moisture, ash, crude protein, crude fat, crude fiber, and total carbohydrate contents were determined according to the method of the AOAC [41]. The content of certain minerals, i.e., manganese (Mn), magnesium (Mg), potassium (K), zinc (Zn), cobalt (Co), and calcium (Ca), were determined and calculated on a dry weight basis.

### 3.3. Preparation of Delonix regia Pigment Extract

The petals of collected *Delonix regia* flowers were rinsed under running water, dehydrated at 50 °C for 3 days, ground, sieved through a 1 mm sieve, packed, and kept at 40 °C. The *Delonix regia* flowers’ flour was steeped in water for three days before being filtered. A rotary evaporator evaporated the solvent in the filtrate; then, the concentrated filtrate was lyophilized. The extract was stored at −20 °C before being analyzed for polyphenols, flavonoids, anthocyanins, and polysaccharides [42].

### 3.4. Polyphenolic Content in DRPE

#### 3.4.1. Total Phenolics (TFs) and Flavonoids (TFs)

As described previously by Wolfe et al. [43], the Folin–Ciocalteu reagent (Sigma, Dokki, Egypt) was employed to evaluate TPs in DRPE. The absorbance of the developed color was estimated at 760 nm. The TPs were represented as mg gallic acid (GA, Sigma, Egypt)/g and evaluated using the linear formula generated from the GA standard curve:y = 0.0226 x + 0.0771, R_2_ = 0.9991(1)

Concerning TFs, they were estimated as quercetin equivalents (QE; Merck Ltd., Cairo, Egypt) by the AlCl_3_ method [44], while the developing color was read at 450 nm depending on the QE standardization curve’s equation:y = 0.0148 x − 0.0107, R_2_ = 0.9996(2)

#### 3.4.2. Anthocyanins

##### Total Anthocyanins (TAs)

The TAs of the flowers were measured following Singh et al.’s [45] pH differential technique. Briefly, 1 g of dried flowers were ground into a powder and extracted in acidic Methanol (HCl 1%). The flask was put in a shaking incubator for six hours at 100 rpm and 25 °C. At 7000 rpm for 10 min, 1 mL of supernatant was added to 20 mL of each pH buffer in separate test tubes after diluting the solution with pH buffers 1.0 and 4.5. The absorbance was read at 520 and 700 nm using pure water as a blank. The anthocyanin pigment content was expressed as cyanidin-3-glucoside equivalents. The previously calculated amounts of HCl and distilled water may differ depending on the experiment. The total anthocyanin content was estimated as cyanidin-3-O-glucoside (Bio-Rad Laboratories, Hercules, CA, USA) in the following equation:(3)$$Totalanthocyanin=\frac{Absorbance\times Molecularwieght\times DF\times 10^{3}}{\varepsilon \times 1}$$
where absorbance is at (A520 nm–A700 nm); *DF* = dilution factor; molecular weight is 449.2 g/mol; ε (molar extinction coefficient) = 26,900 in L mol^–1^ cm^–1^, for cyd-3-glu; and 10^3^ = convert g to mg.

##### Anthocyanins Profile by HPLC

The HPLC (LC-10 AS, Shimazu, Japan) is fully equipped with an auto-sampler, a quaternary pump, a C18 separation column (Gemini, 4.60 mm, 5 μm, 35 °C), a mobile phase flow rate of 1 mL/min, and a multiwavelength detector set at 520 nm to detect anthocyanin compounds following Saad et al. [44].

### 3.5. Polysaccharides Content in Delonix regia Flowers

#### 3.5.1. Extraction of the Crude Polysaccharide

The petals of collected *Delonix regia* flowers were rinsed under running water, dehydrated at 50 °C for 3 days, ground, and sieved through a 1 mm sieve. The lipid was extracted using a Soxhlet apparatus with petroleum ether (b.p. 60–90 °C, Bio-Diagnostic, Egypt) and ethanol (80%) for 3 h. The defatted sample was retained in boiling water (120 °C, 1:30, *w*/*v*). The supernatant was obtained and retained in ethanol (80%) overnight at 4 °C, centrifuged, and then the defatted flour was dried in an oven at 60 °C until the weight remained constant.

The crude polysaccharide was purified using barium complexation by creating a 2.5% (*w*/*v*) solution of the gum by stirring continuously for 12 h at 60 °C and precipitating it with a saturated barium hydroxide solution. Separated by centrifugation, the complex was dissolved in 1 M acetic acid (Merck, Cairo, Egypt), agitated for 8 hours, centrifuged, and precipitated with absolute alcohol. It was cleansed four to five times with pure alcohol. The material was eventually purified by dialysis and filtration using 0.65 m and 0.45 m Millipore membranes following [46].

#### 3.5.2. Monosaccharide Profile

The polysaccharide sample (5 mg) was dissolved in 2 mL of 2 M trifluoracetic acid (TFA, Sigma, Egypt) and hydrolyzed for 3 h at 110 °C in a sealed tube. Under nitrogen flow, the soluble portion was evaporated to dryness. The samples and monosaccharide standards (Bio-Rad, USA) were injected onto a platinum amino column (5 µm, 250 mm 4.6 mm i.d.; Grace, Lokeren, Belgium), which was kept at 20 °C in a column oven following the removal of TFA (CTO-10 ASvp, Shimadzu). The analysis used an LC-10 ADvp pump from HPLC (Shimadzu, Japan) and ELSD (Sedex 55, SEDERE, Olivet, France) at 60 °C and 230 kPa of nitrogen pressure. The chromatograms were obtained, and the data were processed with Class-VP software (Shimadzu, version 6.1, Tokyo, Japan) [47].

#### 3.5.3. Molecular Weight Determination

The molecular weight of the separated polysaccharide was determined due to the method previously described by Ji et al. [48]. by gel permeation chromatography (GPC) on a Sephadex G-150 column (2.6 cm × 70 cm). Standard dextrans (40, 500, and 2000 kDa) and glucose were used, and the elution volumes were plotted against the logarithm of their respective molecular weights. The elution volume of the purified polysaccharide was plotted in the same graph, and the molecular weight was determined by Ji et al. [48].

#### 3.5.4. Characterization of DR Polysaccharides

##### FT-IR Analysis

FT-IR spectra (IS-50) were used to measure the FT-IR spectroscopy of the sample in the region of 400–4000 cm^−1^. An amount of 1 mg of the sample was mixed with 150 mg of dry KBr and then pressed into a pan for analysis [49].

##### ^1^H-NMR Spectroscopy Analysis

Using an AVANCE III 400 MHz NMR spectrometer, ^1^H-NMR spectra were obtained (Bruker Corporation, Switzerland). An amount of 40 mg of a lyophilized sample was dissolved in 0.6 cc D2 O. In ppm, chemical changes were reported [49].

### 3.6. Biological Activities of DRPE

#### 3.6.1. Antioxidant

##### DPPH Assay

The DRPE (0.2, 0.4, 0.8, 1,2, and 1.6 mg/mL) was tested for its ability to scavenge DPPH. First, 100 µL of DRPE was added to 100 µL of DPPH (Sigma, Egypt); the mix was loaded in a microtiter plate and kept in the dark for thirty minutes; the plate was read at 517 nm using a microtiter plate reader (BioTek, Santa Clara, CA, USA), and the obtained values were applied in the equation (Equation (4)).
(4)$$\%Antioxidant\;activity=\frac{Control\;absorbance-sample\;absorbance}{Control\;absorbance}\times 100$$

##### Ferric-Reducing Antioxidant Power (FRAP)

The plant extracts were determined by assessing their ability to decrease FeCl_3_, as described by Moein et al. [50], with minor adjustments. The DRPE concentrations (0.2, 0.4, 0.8, 1.2, and 1.6 mg/mL) were combined with 0.5 mL of a 0.2 M phosphate buffer with a pH of 6.6 and 0.5 mL of K_3_ FeCN_6_ (1%, Bio-Diagnostic, Dokki, Egypt), then incubated for 20 min at 50 °C. Then, TCA (10%) was added, and the mixture was centrifuged for 10 min at 2500 rpm. Then, 0.5 mL of the supernatant was combined with an equal volume of distilled water, and 100 µL of ferric chloride (FeCl_3,_ Merck, Egypt) containing 0.1% ferric chloride was added. The absorbance of the plant extract combination was determined at 700 nm using a spectrophotometer, compared to a blank. Plant extracts with higher values have a greater lowering capacity.

#### 3.6.2. Cytotoxicity Effects

The sulphorhodamine B (SRB; Bio-Diagnostic, Dokki, Egypt) assay was used to evaluate the viability of the pancreatic (Panc-1) cell lines. Cancer cells (5 × 10^3^ cells, Bio-Diagnostic, Egypt) were suspended in 100 µL of control medium and cultured for 24 h in 96-well plates. Under the same circumstances, another 100 µL was added to the medium supplemented with various doses of DRPE (0.2, 0.4, 0.8, 1,2, and 1.6 mg/mL). The samples were treated with 150 µL of 10% TCA (Merck, Egypt) for three days to ensure fixation, incubated for one hour at 4 °C, then rinsed many times with dH_2_ O. The SRB solution (0.4%, 70 µL) was added, and the plates were incubated for 10 min in a dark place. Acetic acid (1%) was used to clean the plates 3 times before being air-dried for an entire night. The protein-bound SRB dye was dissolved in 10 mM of TRIS-HCl (150 µL, Sigma, Egypt); then, the absorbance of the developed color was read at 540 nm using a microtiter plate reader (BioTek Elx808, Santa Clara, CA, USA). The LC_50_ was established as the median concentration that caused a 50% reduction in the absorbance [51].

#### 3.6.3. Antibacterial Activity

DRPE (0.2, 0.4, 0.8, 1, 2, and 1.6 mg/mL) was performed against various bacterial and fungal strains. The bacteria strains were cultivated overnight at 37 °C in a shaking incubator. Mueller–Hinton broth (MHB; Oxoid, Basingstoke, UK) was used at a concentration of (1 × 10^8^ CFU/mL). The disc diffusion method was used to conduct the antibacterial activity [52]. The Petri plates were inoculated using the spread plate technique with 100 µL of active bacterial strains. Previously saturated paper discs (6 mm) with DRPE (0.2, 0.4, 0.8, 1,2, and 1.6 mg/mL) were placed on the plates’ surface. The plates were incubated for 24 h at 37 °C. A ruler estimated the inhibition zones surrounding the discs (in mm). The results were examined with the antibiotic, Levofloxacin, as a positive control. The MIC was estimated following Saad et al.’s study [44].

### 3.7. Safety Experiment

A total of 40 rats (weighing 160–180 g) were obtained from the Faculty of Veterinary Medicine, Zagazig University, Egypt, and allocated randomly into 5 groups of 8 animals. The groups were:The rats were fed a basal diet without additions as a negative control;As a positive control, the rats were fed a basal diet supplemented with a synthetic dye (1.6 mg/mL);As a positive control, the rats were fed a basal diet supplemented with aspartame (1.6 mg/mL);The rats’ group was fed a diet supplemented with DRPE (0.2 mg/g);The rats’ group was fed a diet supplemented with DRPE (1.6 mg/g).

The treatments were given for four weeks. Rats were ethically slaughtered at the end of the trial, and blood samples were obtained from the retro-orbital vein. Biochemical assays of liver enzyme activity, total protein, lipid profile, malonaldehyde, and renal function were performed.

#### Biochemical Examination

ALT, AST, total protein, and albumin were approximated according to Schumann and Klauke [53]. Reduced glutathione in serum was quantified by Beutler’s method [54]. MDA was calculated as the reactive form of TBA (Sigma, USA) according to [54]. Urea and creatinine were measured as markers of renal function [55]. The total cholesterol, LDL, and HDL amounts will be calculated using Armbruster DA Lambert’s enzymatic colorimeter method [56]. Total triglycerides will be determined using Devi and Sharma’s technique [57]. All biochemical kits were purchased from Bio-Diagnostic, Egypt.

### 3.8. Preparation of Strawberry Drink Supplemented with DRPE

The strawberries were rinsed many times with tap water, then juiced by a mixer, following Saad et al.’s method [44] with some modifications. A total of 90 mL of DRPE at different concentrations and 50 g of yogurt were mixed with 250 mL of strawberry juice in a blender. The water was added according to Table 5. The strawberry beverages SB, DRSB1, DRSB2, and DRSB3 were put into 500-mL sterilized bottles at 95 °C for 2 min under 50 MPa in a TOMY Sx-700 autoclave (Japan), then stored at 4 °C.

### 3.9. Color Analysis and Sensory Properties

The color change of ΔE was analyzed using a Hunter Lab spectrophotometer (Color Flex EZ’s, Reston, Virginia, USA) and determined according to Namir et al. [58].

A total of 6 men and 4 women between the ages of 40 and 55 rated the sensory characteristics of yogurt samples. Flavor, color, texture, form, and overall acceptance were assessed. Each participant was given water to remove the effects of each sample, which was then evaluated accordingly [59].

### 3.10. Statistical Evaluation

The statistical analysis experiment was set up in a simple, completely randomized design with 15 replications. For the phytochemical analysis, each treatment was represented by three replicates. The results were analyzed with the analysis of variance (ANOVA) procedure by MSTAT-C Statistical (Michigan State University, 1990). The post hoc test (LSD) was used to compare data means for statistical differences [60].

## 4. Conclusions

Chemical composition constituents of plants *Delonix regia* indicated the presence of carbohydrates and anthocyanins. DRPE has antioxidant, anticancer, and antimicrobial potential, which considerably impacts the sensorial properties of DRPE strawberries, especially color and taste, extending the shelf life and valuing its quality as a functional product. The beneficial properties of DRPE in rats, such as lowering LDL, MDA, ALT, AST, creatinine, and TG while increasing TP and GSH, indicate its safety compared to synthetic dye and aspartame. The worldwide trend is to use natural products as alternatives to synthetic ones to release oxidative stresses caused by chemical additives. Utilizing DRPE as a bioactive supplement and positively reflecting overall health is beneficial. Therefore, adding DRPE to food formulations as a natural colorant and sweetener is recommended for preserving health and the environment.

## Figures and Tables

**Figure 1 molecules-28-03243-f001:**
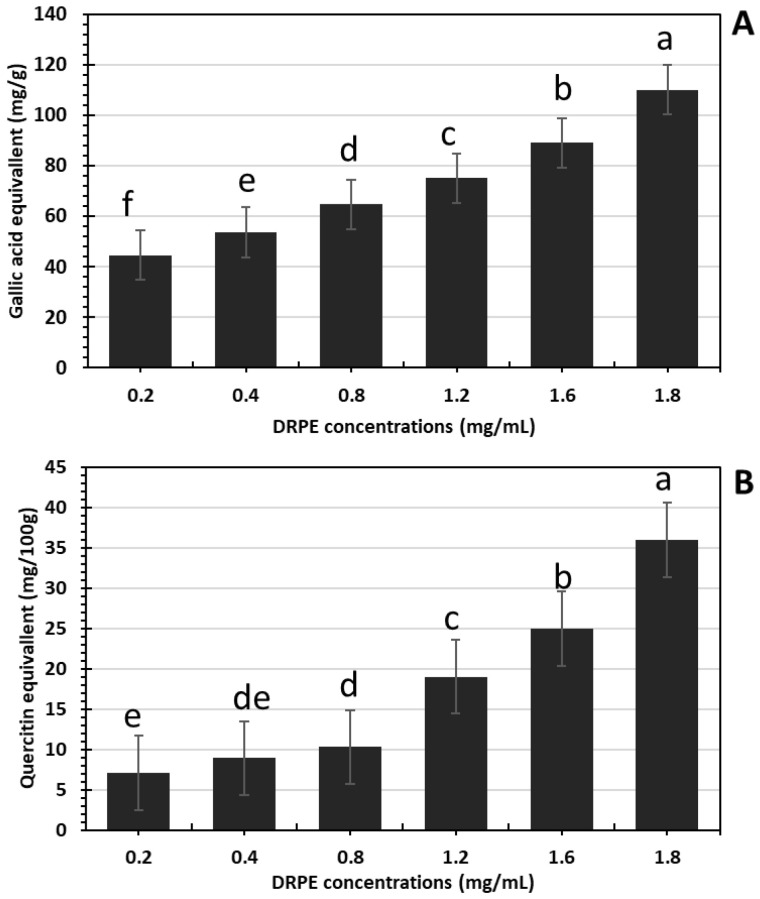
(**A**) Phenolic compounds and (**B**) flavaniods content of DRPE expressed as gallic acid and quercetin equivalents, respectively. Lowercase letters above the columns indicate significant differences.

**Figure 2 molecules-28-03243-f002:**
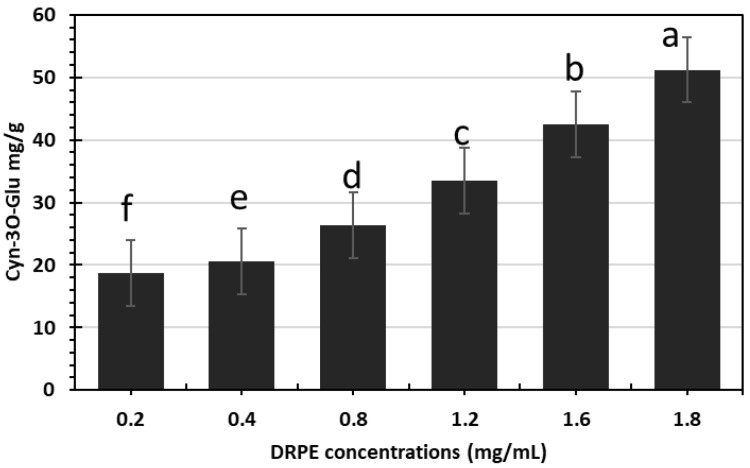
Anthocyanin compounds of DRPE are expressed as cyanidin-3-O-glu equivalents. Lowercase letters above the columns indicate significant differences at a 5% probability level.

**Figure 3 molecules-28-03243-f003:**
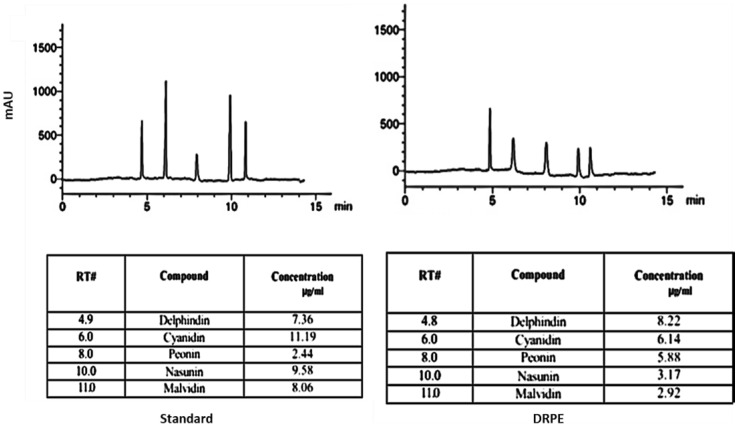
Anthocyanin’s profile detected by HPLC at 520 nm.

**Figure 4 molecules-28-03243-f004:**
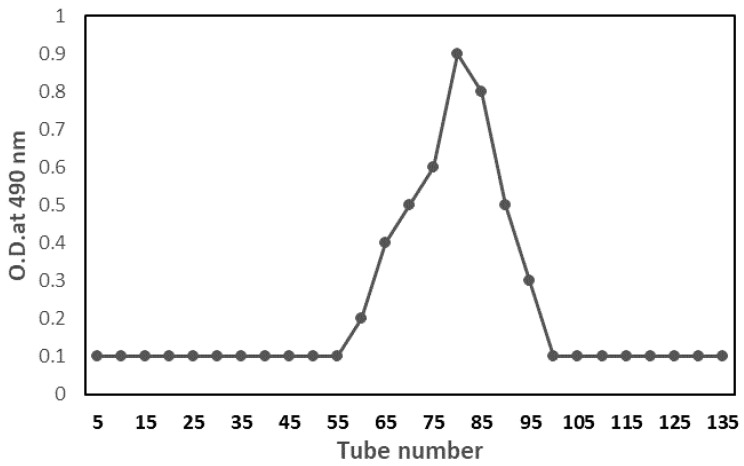
Gel permeation chromatography of polysaccharide fraction on Sephacryl S-200.

**Figure 5 molecules-28-03243-f005:**
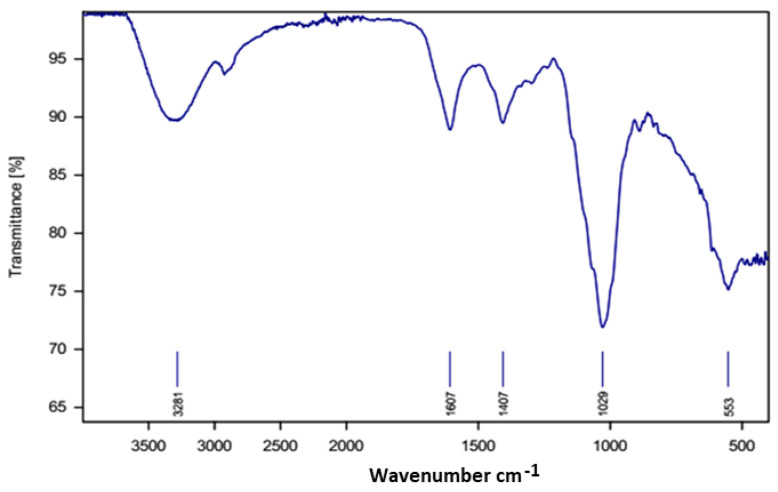
FT-IR spectrum of *Delonix regia* polysaccharide.

**Figure 6 molecules-28-03243-f006:**
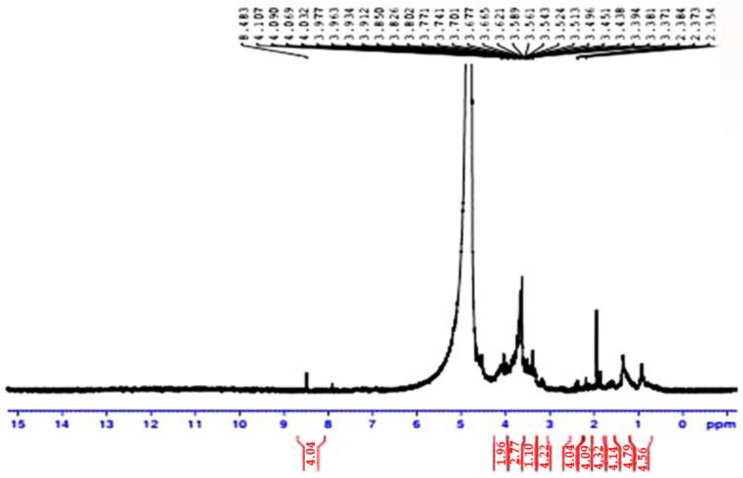
^1^ H NMR spectra show the structure of isolated polysaccharides from *Delonix regia*.

**Figure 7 molecules-28-03243-f007:**
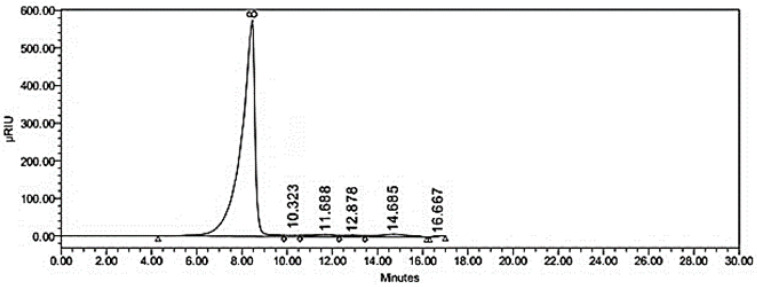
Chromatograms for analysis of the component monosaccharide present in polysaccharide *Delonix regia* using the pre-column PMP derivatization HPLC method.

**Figure 8 molecules-28-03243-f008:**
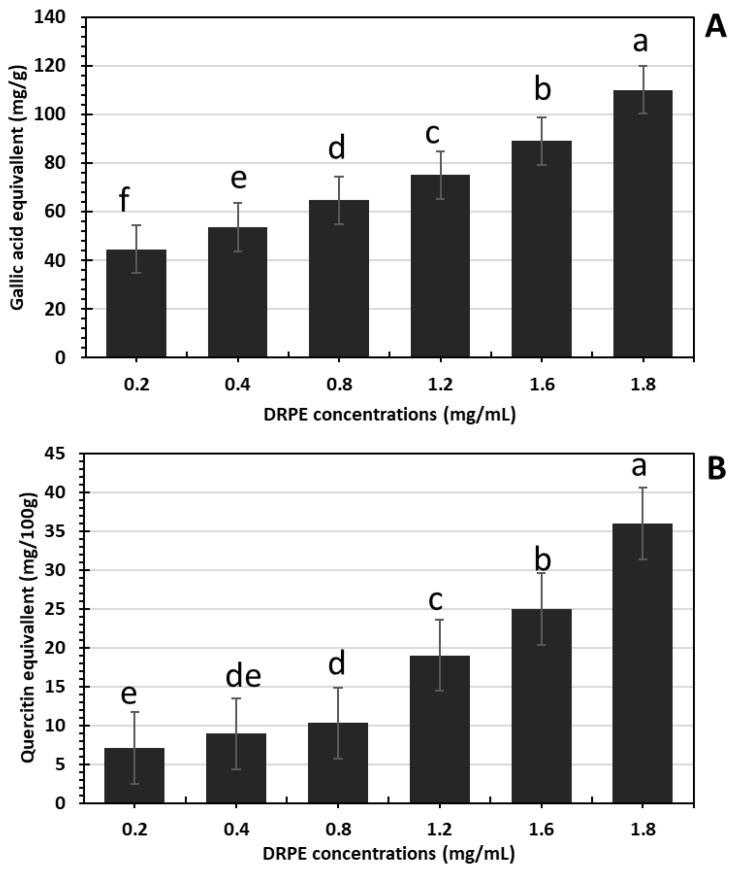
(**A**) Scavenging activity of DRPE concentrations against DPPH free radicals; (**B**) ferric-reducing power of DRPE concentrations. Lowercase letters above the columns indicate significant differences at a 5% probability level.

**Figure 9 molecules-28-03243-f009:**
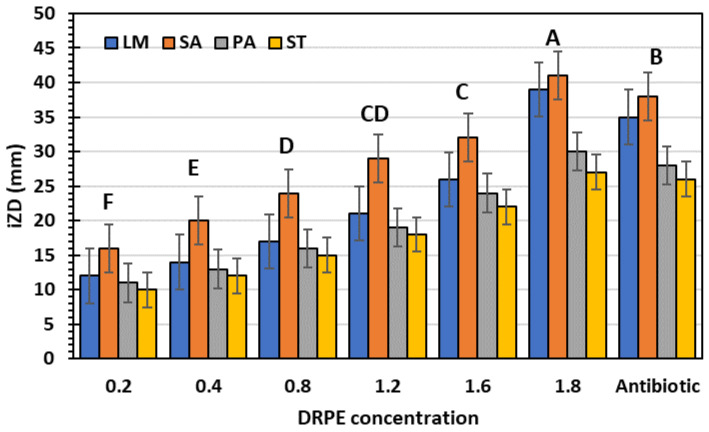
Inhibition zone diameters of DRPE with different concentrations against pathogenic bacteria compared to a bacterial antibiotic after one day of incubation at 37 °C. Uppercase letters above the columns indicate significant differences at a 5% probability level.

**Figure 10 molecules-28-03243-f010:**
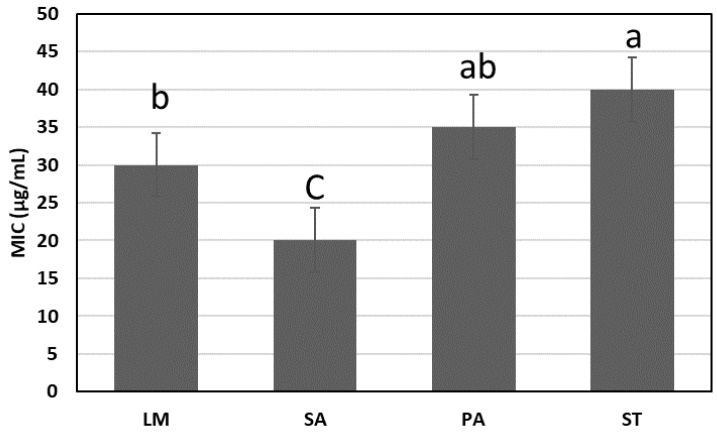
The minimum inhibitory concentration of LPE against tested bacteria. Lowercase letters above the columns indicate significant differences at a 5% probability level.

**Figure 11 molecules-28-03243-f011:**
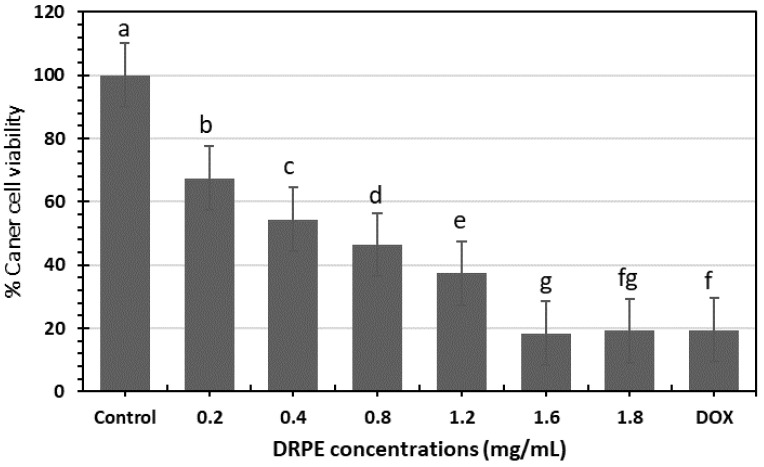
Histogram of the % viability of breast cancer cells affected by DRPE at different concentrations compared to doxorubicin. Lowercase letters above the columns indicate significant differences at a 5% probability level.

**Table 1 molecules-28-03243-t001:** Proximate analysis of the *Delonix regia* flower (on a dry weight basis).

Proximate Composition	Values (%)
Moisture	10.14 ± 0.1 c
Ash	2.05 ± 0.2 d
Protein	10.01 ± 0.1 c
Fiber	14.04 ± 0.9 b
Fat	13.68 ± 0.5 b
Carbohydrates	50.24 ± 0.2 a
Elemental content	Values (ppm)
Mn	13.81 ± 0.1 e
Mg	2220.36 ± 1.8 c
Zn	671.95 ± 2.1 d
Co	0.0125 ± 0.0002 g
K	16,358.12 ± 3.3 a
P	2.087 ± 0.1 f
Ca	4617.15 ± 4.3 b

Data are expressed as mean± standard error of three replicates. Lowercase letters in the column indicate significant differences.

**Table 2 molecules-28-03243-t002:** Monosaccharide profile of isolated DR polysaccharides detected by HPLC.

RT	Monosaccharide	Concentration (µM)
8.00	Mannose	3.60 a
10.32	Glucose	1.00 c
12.78	Galactose	2.70 b
14.68	Xylose	0.78 d
16.66	Arabinose	1.10 c

Lowercase letters in the column indicate a significant difference.

**Table 3 molecules-28-03243-t003:** Serum parameters of rats affected by synthetic dye, aspartame, and DRPE (µg/g).

Blood Parameters	Control	Synthetic Dye(µg/g)	Aspartame (µg/g)	DRPE (µg/g)	
Liver parameters		1600	1600	200	1600
ALT	45.9 ± 0.2 b	89.6 ± 0.9 a	85.4 ± 0.2 ab	47.2 ± 0.2 b	46.5 ± 0.3 b
AST	32.6 ± 0.3 c	81.2 ± 0.3 a	77.6 ± 0.1 b	33.9 ± 0.3 c	33.6 ± 0.7 c
MDA	46.3 ± 0.7 c	71.3 ± 0.5 a	69.2 ± 0.3 ab	47.2 ± 0.1 c	46.9 ± 0.9 c
GSH	55.6 ± 0.9 ab	41.6 ± 0.8 b	39.6 ± 0.9 b	55.9 ± 0.6 ab	56.2 ± 0.8 a
Kidney parameters					
Urea					
Creatinine	0.41 ± 0.01 b	0.89 ± 0.02 a	0.88 ± 0.03 a	0.46 ± 0.01 b	0.45 ± 0.09 b
Lipid profile					
LDL	16.8 ± 0.2 c	87.3 ± 0.6 a	85.6 ± 0.2 ab	18.6 ± 0.9 b	17.1 ± 0.2 c
HDL	35.2 ± 0.9 b	21.7 ± 0.1 c	22.6 ± 0.1 c	36.2 ± 0.1 b	41.3 ± 0.3 a
TG	72.3 ± 0.8 c	140.3 ± 0.2 a	133.5 ± 0.6 b	73.9 ± 0.2 c	71.2 ± 0.8 c
TC	66.9 ± 0.2 c	126.4 ± 0.9 a	121.9 ± 0.5 b	68.2 ± 0.7 c	66.3 ± 0.4 c

Data are presented as mean ± SD. Lowercase letters within raw indicate significant differences at the 5% level.

**Table 4 molecules-28-03243-t004:** Sensorial properties of DR strawberry beverages (200, 800, and 1600 µg/g) during cold preservation (mean ± SD).

Yogurt Samples		Storage (Days)	Color	Flavor	Texture	Taste	Overall Acceptability
DR concentration (µg/g)	Control (0)	0	9.0 ± 0.0 a	9.0 ± 0.0 a	8.2 ± 0.0 cd	8.4 ± 0.2 a	8.7 ± 0.0 bc
		7	8.7 ± 0.2 bc	8.4 ± 0.1 c	8.2 ± 0.1 cd	8.1 ± 0.1 bc	8.4 ± 0.1 cd
		14	8.5 ± 0.3 cd	8.1 ± 0.0 d	8.0 ± 0.2 d	7.9 ± 0.0 c	8.2 ± 0.2 d
		21	8.0 ± 0.2 e	7.6 ± 0.2 e	7.5 ± 0.1 e	7.4 ± 0.1 e	7.7 ± 0.3 e
		30	7.5 ± 0.7 f	7.5 ± 0.3 e	7.0 ± 0.0 f	6.9 ± 0.2 f	7.3 ± 0.4 f
	200	0	9.0 ± 0.0 a	9.0 ± 0.0 a	8.5 ± 0.2 c	8.4 ± 0.4 a	8.8 ± 0.1 b
		7	8.7 ± 0.1 bc	8.5 ± 0.2 c	8.4 ± 0.1 c	8.1 ± 0.2 bc	8.5 ± 0.2 cd
		14	8.6 ± 0.2 c	8.3 ± 0.3 cd	8.2 ± 0.3 cd	8.0 ± 0.1 c	8.4 ± 0.1 d
		21	8.2 ± 0.3 d	8.1 ± 0.2 d	8.0 ± 0.2 d	7.6 ± 0.1 d	8.1 ± 0.0 de
		30	8.1 ± 0.2	8.0 ± 0.1 d	7.6 ± 0.1 f	7.5 ± 0.2 de	8.0 ± 0.2 e
	800	0	9.0 ± 0.0 a	9.0 ± 0.0 a	8.9 ± 0.0 a	8.4 ± 0.0 a	9.0 ± 0.0 a
		7	8.8 ± 0.0 b	8.8 ± 0.1 b	8.7 ± 0.1 b	8.2 ± 0.2 b	8.8 ± 0.1 b
		14	8.6 ± 0.1 c	8.7 ± 0.2 bc	8.5 ± 0.2 c	8.1 ± 0.1 bc	8.6 ± 0.2 c
		21	8.6 ± 0.1 c	8.5 ± 0.1 c	8.3 ± 0.1 cd	8.0 ± 0.1 c	8.5 ± 0.1 cd
		30	8.5 ± 0.2 cd	8.3 ± 0.2 cd	8.0 ± 0.0 d	7.9 ± 0.1 cd	8.3 ± 0.2 d
	1600	0	9.0 ± 0.0 a	9.0 ± 0.0 a	8.6 ± 0.0 bc	8.4 ± 0.1 a	8.9 ± 0.0 ab
		7	8.8 ± 0.1 b	8.7 ± 0.2 bc	8.5 ± 0.1 c	8.2 ± 0.2 b	8.7 ± 0.1 bc
		14	8.7 ± 0.2 bc	8.5 ± 0.1 c	8.3 ± 0.2 cd	8.1 ± 0.1 bc	8.5 ± 0.0 cd
		21	8.5 ± 0.1 cd	8.4 ± 0.2 c	8.0 ± 0.2 d	7.9 ± 0.2 c	8.3 ± 0.2 d
		30	8.3 ± 0.1 d	8.2 ± 0.1 cd	7.7 ± 0.1 f	7.7 ± 0.1 d	8.1 ± 0.1 de

Lowercase letters in each column indicate a significant difference between samples at a probability level of 5%.

**Table 5 molecules-28-03243-t005:** The ingredients of DRPE-enriched strawberry beverage.

Ingredients	SB	DRSB1	DRSB2	DRSB3
Juice (mL)	250	250	250	250
yogurt (g)	50	50	50	50
Sugar (g)	10	-	-	-
DRPE (mL)	-	90	90	90
Water (mL)	90	-	-	-

Strawberry beverage (SB); control (C); strawberry beverage supplemented with DRPE 200 µg/mL (DRSB1); strawberry beverage supplemented with DRPE 800 µg/mL (DRSB2); strawberry beverage supplemented with DRPE 1600 µg/mL (DRSB3).

## Data Availability

Not applicable.

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
