# Peer review of "Characterization of Delonix regia Flowers’ Pigment and Polysaccharides: Evaluating Their Antibacterial, Anticancer, and Antioxidant Activities and Their Application as a Natural Colorant and Sweetener in Beverages"

_molecules, 2023, doi:10.3390/molecules28073243_

Round 1
Reviewer 1 Report
There is an interesting article, however, I have some questions for the authors.
1. Have any side effects from Delonix regia flowers?
2. What kind of colors are from Delonix regia flowers or crude Delonix regia flower polysaccharides?
Thank you very much.
Author Response
Reviewer 1 comments
Comments and Suggestions for Authors
There is an interesting article, however, I have some questions for the authors.
Response: Thanks very much for the positive impact of the reviewer on our manuscript
1.Have any side effects from Delonix regia flowers?
Response: Acutally, we tested the safety of DR flower extract in rats' diets compared to synthetic dye and aspartame; we found regular liver markers (AST, ALP, and ALT), kidney markers (urea and creatinine), high TP, TA, and GSH, and low MDA, confirming the safety of DRPE. Also previous studies support these results Rahman et al. (2020) found that flower extracts from Delonix regia were rich in phytochemicals and potential bioactivities, which indicates the possibility of these flowers being used as a source of phytochemicals as well as safe and effective natural medicine. Furthermore, El-Sayed et al. (2011) suggested using the ethanolic extract of the flowers of D. regia as a chemopreventive agent against the two main causes of liver damage; liver toxicity by chlorinated agents and liver cancer. Also, The alcoholic extract of the Delonix regia Linn flowers was evaluated for antiarthritic activity in adult female Wistar rats, where it safely reduced paw edema volume (Chitra et al., 2010). Moreover, Morsi et al. (2019) found that the carotenoid-rich extract of DR flower can be used as a bio-preservative in sunflower oil for stability against oxidation.
References
Chitra, V., Ilango, K., Rajanandh, M.G., Soni, D., 2010. Evaluation of Delonix regia Linn. flowers for antiarthritic and antioxidant activity in female wistar rats. Annals of Biological Research, 1(2), 142-147..
El-Sayed, A.M., Ezzat, S.M., Salama, M.M., Sleem, A.A., 2011. Hepatoprotective and cytotoxic activities of Delonix regia flower extracts. Pharmacognosy Journal, 3(19), 49-56..
Morsi, M.K.S., Morsy, N.F.S., Golshany, H.S., 2019. Efficiency of ultrasound assisted extract of Delonix regia petals as natural antioxidant on the oxidative stability of sunflower oil. Grasas y Aceites, 70(4), e332-e332..
Rahman, F.B., Ahmed, S., Noor, P., Rahman, M.M.M., Huq, S.M.A., Akib, M.T.E., Shohael, A.M., 2020. A comprehensive multi-directional exploration of phytochemicals and bioactivities of flower extracts from Delonix regia (Bojer ex Hook.) Raf., Cassia fistula L. and Lagerstroemia speciosa L. Biochemistry and Biophysics Reports 24, 100805.
- What kind of colors are from Delonix regia flowers or crude Delonix regia flower polysaccharides?
Response: The anthocyanin-rich extract of DR flowers is the source of the red color, so we use it as a colorant in strawberry beverages to enrich the color. Another part of the flower extract was used to extract polysaccharides to apply as a sweetener
Thank you very much.
Response: Thanks very much for the positive impact of the reviewer on our manuscript
Reviewer 2 Report
Dear authors,
I found your manuscript very interesting and scientifically well presented. The research goal was to investigate the potential use of Delonix regia flower, the flame tree, in beverages. The introduction provides sufficient information. The results showed that Delonix regia pigments could be used as natural colorant and sweetener, especially due to its antibacterial, anticancer and antioxidant activities. This is important, because the latest studies are focused on substituting the synthetic sweeteners and colorants.
This manuscript contains large amount of results, and all are appropriately explained and discussed, and accompanied with appropriate references.
Tables and figures are clear and easy to read and understand.
Methods are well described. I only recommend adding what chemicals were used and where they were purchased.
The Abstract is a slightly longer, but considering the research size, it is understandable.
Please check the text for some technical errors. For example, between the temperature number and Celsius sign goes a space, like in line 129, and unlike in line 130, same sentence. Same thing appears through text.
In line 200, please write number "-1" in cm-1 as exponent. Check the rest of the text.
In line 250. there should be space between number and gram unit. Check the rest of the text.
Other than that, I do not have anymore comments and it is my opinion that this manuscript could be published in this journal.
Author Response
Reviewer 2 comments
Comments and Suggestions for Authors
Dear authors,
I found your manuscript very interesting and scientifically well presented. The research goal was to investigate the potential use of Delonix regia flower, the flame tree, in beverages. The introduction provides sufficient information. The results showed that Delonix regia pigments could be used as natural colorant and sweetener, especially due to its antibacterial, anticancer and antioxidant activities. This is important, because the latest studies are focused on substituting the synthetic sweeteners and colorants.
Response: Thanks very much for the positive impact of the reviewer on our manuscript
This manuscript contains large amount of results, and all are appropriately explained and discussed, and accompanied with appropriate references.
Response: Thanks very much for the positive impact of the reviewer on our manuscript
Tables and figures are clear and easy to read and understand.
Response: Thanks very much for the positive impact of the reviewer on our manuscript
Methods are well described. I only recommend adding what chemicals were used and where they were purchased.
Response: Done as requested
The Abstract is a slightly longer, but considering the research size, it is understandable.
Response: Thanks very much for the positive impact of the reviewer on our manuscript
Please check the text for some technical errors. For example, between the temperature number and Celsius sign goes a space, like in line 129, and unlike in line 130, same sentence. Same thing appears through text.
Response: Done as requested
In line 200, please write number "-1" in cm-1 as exponent. Check the rest of the text.
Response: Done as requested
In line 250. there should be space between number and gram unit. Check the rest of the text.
Response: Done as requested
Other than that, I do not have anymore comments and it is my opinion that this manuscript could be published in this journal.
Response: We appreciate the positive comments of the reviewer. Thanks very much